# Impact of Exogenous Factors and Anesthetic Risk in Premature Birth during the Pandemic Period

**DOI:** 10.3390/diagnostics14111123

**Published:** 2024-05-29

**Authors:** Florin Tovirnac, Carolina Susanu, Nicoleta Andreea Tovirnac, Eva Maria Elkan, Ana Maria Cobzaru, Alexandru Nechifor, Alina Mihaela Calin

**Affiliations:** 1Clinic Surgical Department, Faculty of Medicine and Pharmacy, Dunarea de Jos University of Galati, 800008 Galati, Romania; tovarnacf@yahoo.com (F.T.);; 2Morphological and Functional Sciences Department, Faculty of Medicine and Pharmacy, Dunarea de Jos Unversity of Galati, 800008 Galati, Romania; 3Clinical Department, Faculty of Medicine and Pharmacy, Dunarea de Jos University of Galati, 800008 Galati, Romania; alexandru.nechifor@ugal.ro

**Keywords:** premature birth, anesthetic risk, alcohol, smoking, prematurity

## Abstract

Background: Premature birth remains a public health problem worldwide, involving a broader context and a multidisciplinary team aimed at combating this phenomenon as much as possible. The consumption of addictive substances by women who are pregnant can occur in different social contexts and at different stages of their lives, which modulate its extent. Obstetricians and anesthetists should consider the anesthetic maternal risks that may arise due to these addictive behaviors. The maternal anesthetic risk is higher in women who are pregnant with a medium-level of education, imbalanced nutrition, stress associated with physical or mental activity, affected sleep hygiene, and failed marriages. Objectives: The objectives of the study refer to analyzing the impact of exogenous factors and the anesthetic risk on premature birth for women who were pregnant during the pandemic period and in women who were pregnant without COVID-19 infection. The authors studied a significant sample of 3588 women who were pregnant without COVID-19 infection, among whom 3291 gave birth at term and 297 gave birth prematurely. Methods: The methods analyzed consist of studying the specialized literature regarding the impact of exogenous factors and parturient’s anesthetic risk on premature birth and identifying the regional risk profile of women who are pregnant in the southeast region of Romania compared to that identified in the specialized literature. In the analytical methods, we used a linear regression to study the incidence of exogenous risk factors on anesthetic risk in women who were pregnant with premature births compared to those with full-term births. Results: The results confirm the significant impact of exogenous factors on anesthetic risk and the significant impact of anesthetic risk on premature births. The novelty of the study lies in highlighting the modification of the regional exogenous risk profile during the pandemic period in southeast Romania due to unfavorable socio-economic causes and the translation of grade I and II prematurity events to higher frequencies with an increased level of maternal anesthetic risk. Conclusions: The study findings show that the anesthetic risk is maximized in parturients with a middle school education. Additionally, the anesthetic risk of patients who are pregnant increases with the intensification of smoking adherence and its maintenance throughout the pregnancy at the same intensity. Our study aims to provide a basis for the diversification and development of community intervention programs in the post-COVID-19 era, considering the reshaping of social models and the repositioning of social principles and values. Obstetricians and anesthetists must know and promote family values to harmonize the lives of family members and provide a better life for the mother and child.

## 1. Introduction

Nowadays, premature birth is linked to the lifestyle of mothers [1], stress, access to medical services throughout pregnancy, beliefs, customs and values of the mother’s family, and her social environment [2]. A public health issue includes old addictions such as alcohol [3,4,5,6], smoking [7,8], and newer prohibited substances [9], as well as many medications (benzodiazepines, herbal medicines, and atypical antidepressants) [10]. The parturient’s anesthetic risk in premature birth [11,12,13,14,15] may be higher than in the case of a full-term birth due to the immaturity of the child’s nervous system and other factors related to prematurity [16], such as the pregnant woman’s health condition, exogenous factors, mental status, and socio-economic and emotional conditions. However, anesthesia is often necessary to control pain [15,17] and allow doctors to surgically intervene in specific obstetric indications and anesthetic risks [18,19,20] of the patient that may arise during premature birth. A key factor in resolving cases is collaboration within the medical team, which must quickly assess all the risks and benefits of anesthesia in these situations and choose the safest and most effective method of pain control for both the mother and the child [21,22]. Knowledge of regional risk profiles induced by exogenous factors and the incidence of maternal anesthetic risk in the population of the region can provide valuable information that can shorten the case analysis times and minimize the risk for both the mother and the fetus [23]. Careful monitoring and a cautious approach to risks may be essential for both mother and [24], with the mother needing to consent to the surgical intervention.

The objectives of this study are as follows:Objective 1: Study of specialized literature regarding the impact of exogenous factors and maternal anesthetic risk in premature birth to identify differences between the regional risk profile of women who are pregnant in the southeast region of Romania compared to that identified in the specialized literature.Objective 2: Consolidation of a database for observational study, determining the sample structure.Objective 3: Development of the anesthetic risk model for the two groups of women who were pregnant with term births and women who are pregnant with premature births.Objective 4: Development of public health policies to mitigate the influence of exogenous factors on parturient’s anesthetic risk in premature birth.

The study continues with the presentation of the literature review analysis, the presentation of the main methodological approaches, results, and discussions aimed at building a program of public policies to reduce the influence of analyzed exogenous factors on the mother’s anesthetic risk in premature birth.

## 2. Methodology

We aimed to create a regional intervention plan for reducing the parturient’s anesthetic risk (with the necessity of developing a public awareness program regarding the influence of these studied exogenous factors with a predictive value in premature births).

Data were retrospectively collected based on the observation sheets of women who were pregnant and who gave birth at the Emergency County Clinical Hospital of Brăila, building D, during the pandemic years 2020–2021. The patients in the experimental group (EL) and the control group (CL) are women who were pregnant without COVID-19 infection at the time of birth and breastfeeding. The patients were divided into age groups, residential areas, and education levels, as well as based on the analysis of addictive behaviors, smoking adherence, alcohol adherence, and cumulative adherence. It was analyzed whether the births were normal or with anesthesia, as well as the weight of the children, which were divided into the 4 classes of prematurity recognized by WHO: prematurity grade I with a birth weight between 2000 and 2500 g, prematurity grade II with a birth weight between 1500 and 2000 g, prematurity grade III with a birth weight between 1000 and 1500 g, prematurity grade IV with a birth weight < 1000 g. The patients were analyzed by age groups who sought to give birth at the Emergency County Clinical Hospital of Brăila, building D (see Table 1).

The working hypotheses defined in the study are as follows:

**Hypothesis** **1:**
*Adherence to smoking and alcohol consumption are exogenous risk factors for the parturient’s anesthetic risk.*


**Hypothesis** **2:**
*Women who give birth prematurely—newborns with a low birth weight—exhibit a higher maternal anesthetic risk compared to women who give birth at term with normal weight infants.*


**Hypothesis** **3:**
*In the case of premature births, the risk of exogenous factors represents only one component inducing anesthetic risk for the patient. This risk is also determined by the mother’s genetic structure, history of prematurity, twin pregnancies, and conditions occurring during pregnancy.*


The parturient’s anesthetic risk model for the two groups was developed using multiple linear regressions, with the regression coefficients determined using the least squares method.
(1)$$\begin{array}{ccc} A_{CL}=0.872 & \times & {Age\;}_{CL}-0.334\times {ResidentialArea\;}_{CL}-0.262\times {Studies\;}_{CL}\\ & & +0.346\times {SmokingAdherence}_{CL}\\ & & -0.223\times {AlcoholAdherence}_{CL}\\ & & -0.006\times {AdherenceToVices}_{CL}-0.042\times {BirthWeight}_{CL}\\ & & +0.675 \end{array}$$
(2)$$\begin{array}{ccc} A_{EL}=0.446 & \times & {Age\;}_{EL}-0.023\times {\;ResidentialArea}_{EL}-0.356\times {Studies\;}_{EL}\\ & & +0.131\times {SmokingAdherence}_{EL}\\ & & +0.114\times {AlcoholAdherence}_{EL}\\ & & +0.037\times {AdherenceToVices}_{EL}\\ & & +0.126\times {BirthWeight}_{EL}+0.632 \end{array}$$

From the equations of the model, we draw the conclusion that the age of the patient has a greater influence in the case of the control group than in the case of the experimental group, a fact motivated by the different size of the observations of the two groups, the experimental groups having an episodic nature (during the pandemic period) compared to the general phenomenon of childbirth. It is observed from the equations of the model that there is no difference in sign in both cases, the age varying directly proportional to the maternal anesthetic risk (the mother’s anesthetic risk increases as the pregnant woman advances in age). Both equations in the model demonstrate that there is an inversely proportional correlation between residential status and maternal anesthetic risk, meaning that urbanization has a direct effect on the increase of a mother’s anesthetic risk, significantly higher in the case of the control group than in the case of the experimental group, which demonstrates that for women who are pregnant and exposed to anesthetic risk, there are other influencing factors that determine this risk more comprehensively. From the equations of the model, we draw the conclusion that there is a direct correlation between smoking adherence and maternal anesthetic risk. This aspect is highlighted in both equations for the control group and the experimental group, with the mention that in the case of the experimental group, the correlation coefficient value is lower than in the case of the control group because smoking adherence is not the only exogenous factor with an impact on women who are pregnant with anesthetic risk. Another significant correlation coefficient in the treatment group is represented by alcohol adherence, thus, unlike the control group, this manifests a direct correlation with parturient’s anesthetic risk and adds risk together with smoking adherence. In the case of the control group, an inversely proportional correlation of alcohol adherence is observed, with the effect of risk being reduced. This observation is also formulated for adherence to vices, which, in the case of the experimental group, is directly proportional to maternal anesthetic risk, contrary to the observations for the control group. Therefore, for the experimental group, Hypothesis 1 is demonstrated in that smoking and alcohol adherence are exogenous risk factors for a mother’s anesthetic risk. 

The last factor analyzed refers to birth weight, whose direct correlation is maximized in the case of the experimental group, thus demonstrating Working Hypothesis 2: women who give birth prematurely—newborns under normal weight—exhibit a higher anesthetic risk than women who give birth to full-term babies with normal weight.

## 3. Results 

The model was tested econometrically, with Table 2 showing that the level of statistical representativeness falls within average values for the experimental group and high values for the control group. The dependent variable is characterized by the variation of independent variables by 55% in the case of the experimental group and 75% in the case of the control group. The standard error value of the estimator is higher in the experimental group than in the control group.

The table presented is a statistical summary for regression analysis, which evaluates the influence of independent variables (Birth Weight, Age, AddictionAdherence, Education, SmokingAdherence, ResidentialArea, AlcoholAdherence) on the dependent variable (Anesthetic Risk) across batches. R square change is 0.758 for the control group and 0.562 for the experimental group, indicating that adding predictor variables to the model significantly improved model fit. F change is 1472.827 for the control group and 52.995 for the experimental group, indicating that the variables significantly enhance the model. The analysis shows that both models have a strong correlation and are capable of explaining a significant proportion of the variation in the dependent variable, maternal anesthetic risk, based on the selected independent variables. In the case of the control group model, statistical representativeness is superior to the proposed model for the experimental group because significant influences exist in the experimental group due to factors other than exogenous factors for the manifestation of the patient’s anesthetic risk. The results suggest the importance and relevance of the selected variables in predicting maternal anesthetic risk, with potential implications for medical research or clinical interventions. From Table 2, it is observed that the significance level of errors in both cases is below the 5% error representation threshold, thus validating both models. The null hypothesis is rejected, and the alternative hypothesis is accepted.

It is observed that both models demonstrate a statistically significant capacity to predict the anesthetic risk of women who are pregnant, but the experimental batch model does not capture all aspects that determine the parturient’s anesthetic risk, with specific test values being significantly lower than those in the control batch, which demonstrates hypothesis 3. In the case of premature births, the risk of exogenous factors represents only a component that induces maternal anesthetic risk, which is also determined by the mother’s genetic structure, history of prematurity, twin pregnancies, and pregnancy-related conditions.

## 4. Discussion 

The pyramid distribution of anesthetic risk in pregnant women in relation to exogenous factors is presented in Figure 1.

From Figure 1, it is obvious that residential area has a significant influence on the parturient’s anesthetic risk, with an inverse correlation suggesting that urban living conditions, pollution, and stress increase the risk of anesthesia. The diagram illustrating the association between a mother’s anesthetic risk and residential area highlights higher correlations in both groups within the urban risk segment. In terms of age, maternal anesthetic risk is associated with younger ages between 19 and 39 years old. Education shows a strong negative correlation with the pregnant patient’s anesthetic risk in both groups, indicating that a higher level of education is associated with a healthier lifestyle or better risk management for health, which reduces maternal anesthetic risk.

The results indicate that lifestyle and level of education have a significant impact on health and the need for medical interventions involving anesthesia. These findings underscore the importance of promoting healthy behaviors and education to reduce health risks. A comparative analysis of the maternal anesthetic risk table for the two groups is presented in Table 3 below.

The anesthetic risk of women who are pregnant is influenced by a complex network of interconnected factors, including residential area, level of education, and adherence to risky behaviors, such as smoking and alcohol consumption. The pandemic has been a challenging time in many areas, including antenatal and obstetric care. The pandemic period marked a significant increase in stress and anxiety levels among women who are pregnant, fostering tobacco and alcohol use amidst health uncertainties and limited access to antenatal care. This exogenous stress has been correlated with an increased rate of preterm birth, highlighting the importance of psychosocial support for women who are pregnant. The pandemic period has brought into focus the complexity of managing preterm birth, emphasizing the need for a holistic approach that takes into account both exogenous factors and anesthetic risks in women who are pregnant. In the study, we demonstrated that urban residential areas are associated with increased maternal anesthetic risk due to environmental factors such as pollution and noxious substances. This finding highlights the importance of collaboration between different authorities and sectors to address these risk factors and implement environmental and urban planning solutions that contribute to reducing a parturient’s anesthetic risk.

## 5. Conclusions

This research aimed to determine the impact of exogenous factors and parturient’s anesthetic risk on preterm birth during the pandemic period. The study achieved its four objectives. 

The study’s limitations are primarily due to its regional focus, with data exclusively collected from the south-eastern region of Romania. This restricts the generalizability of the findings to other populations with different socio-demographic characteristics. Additionally, data on substance use and stress levels may be biased due to the self-reporting nature of the research. The authors suggest expanding the research across Europe, potentially through extensive collaboration with doctors from other European countries, to refine the conclusions and enhance generalizability. 

In conclusion, to reduce maternal anesthetic risk in pandemic and post-pandemic contexts, a multidisciplinary approach is essential, including collaboration between health authorities, education, and the environment, as well as awareness and education campaigns tailored to community needs.

## Figures and Tables

**Figure 1 diagnostics-14-01123-f001:**
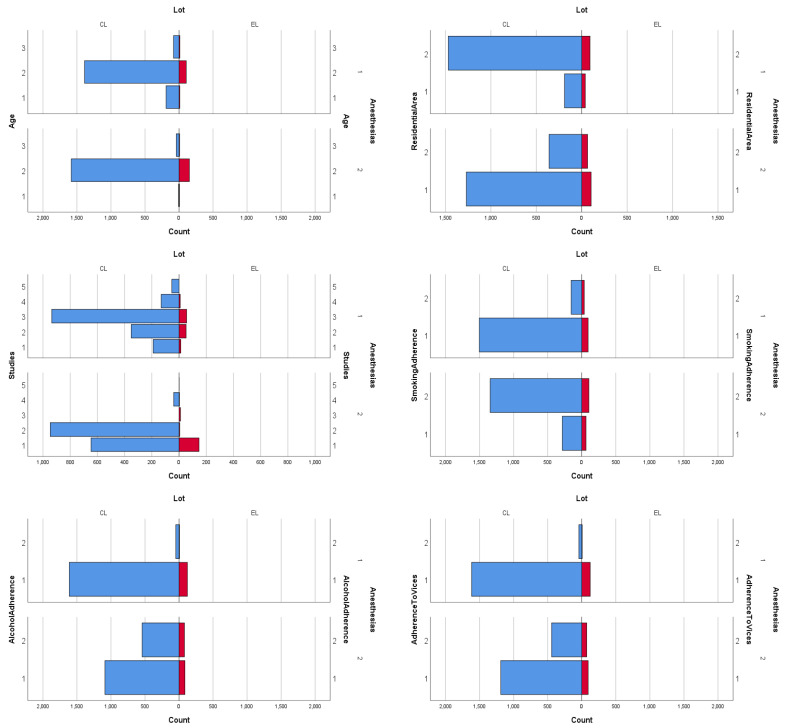
Pyramid distribution of anesthetic risk in relation to exogenous factors (Blue: CL, Red: EL, Anesthesias (1: No/2: Yes), Age (1: 12–18 years, 2: 19–39 years, 3: >40 years), ResidentialArea (1: Urban/2: Rural), Studies (1: Gymnasium, 2: High-school studies, 3: Higher education, 4: Without studies, 5: Not specified), SmokingAdherence (1: No/2: Yes), AlcoholAdherence (1: No/2: Yes), AdherenceToVices (1: No/2: Yes). Source: Created by the author using IBM SPSS software 26 version, International Business Machines Corp, Deutschland GmbH D-71137 Ehningen, Germany.

**Table 1 diagnostics-14-01123-t001:** Regional risk profile of women who are pregnant in the southeast region of Romania (control group CL and experimental group EL).

Type	Indicators
Structural characteristics	Age (3 level)
	ResidentialArea (Urban/Rural)
	Studies (5 levels)
Exogen’s factors	SmokingAdherence (No/Yes)
	AlcoholAdherence (No/Yes)
	AdherenceToVices (No/Yes)
Obstetrics factors	Anesthesias (No/Yes)
	BirthWeight (5 levels)

Source: Created by the author.

**Table 2 diagnostics-14-01123-t002:** Summary model.

Model ^c^	R		R Square	Adjusted R Square	Std. Error of the Estimate	Change Statistics				
	Lot (Selected)	Lot ^b^ (Unselected)				R Square Change	F Change	df1	df2	Sig. F Change
CL	0.871 ^a^	0.542	0.758	0.758	0.246	0.758	1472.827	7	3283	0.000
EL	0.750 ^a^	0.744	0.562	0.551	0.333	0.562	52.995	7	289	0.000

Source: Created by the author. a. Predictors: (Constant), BirthWeight, Age, AdherenceToVices, Studies, SmokingAdherence, ResidentialArea, AlcoholAdherence. b. Statistics are based only on cases for which Lot is CL or EL. c. Dependent Variable: Anesthesias.

**Table 3 diagnostics-14-01123-t003:** The comparative analysis of the anesthetic risk table for the two batches.

Control Group CL	Experimental Group EL
Findings	
The maternal anesthetic risk is inversely proportional to the residential area. If the residential area is urban, the mother’s anesthetic risk increases.	The patient’s anesthetic risk is inversely proportional to the residential area. A decrease in the influence of residential status, compared to the control group, by 50% is observed.
A lower level of education increases the maternal anesthetic risk.	Environmental pollution and stress are no longer the main reasons for a mother’s anesthetic risk; instead, the lack of education is being emphasized.
If smoking adherence increases by 100%, then the maternal anesthetic risk increases by 70%. Alcohol adherence is associated with smoking adherence in 50% of cases.	Smoking adherence becomes the main factor of maternal anesthetic risk, being inversely correlated with residential status, and strongly correlated with alcohol adherence and with both vices, resulting in a risk of 63%.

Source: Created by the author.

## Data Availability

The data that support the findings of this study are available from the corresponding author upon request.

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
