# Peer review of "Impact of Exogenous Factors and Anesthetic Risk in Premature Birth during the Pandemic Period"

_diagnostics, 2024, doi:10.3390/diagnostics14111123_

Round 1

Reviewer 1 Report

Comments and Suggestions for Authors

The article investigates the influence of exogenous factors and anesthetic risks associated with premature births during the COVID-19 pandemic. The study focuses on a significant sample of 3,588 non-COVID pregnant women in Southeast Romania, among whom 297 experienced premature births. The authors analyze the impact of factors such as substance use, stress, inadequate nutrition, and socio-economic conditions on anesthetic risk and the incidence of premature birth.

The study highlights that substance use, including alcohol, tobacco, and other drugs, increases anesthetic risks in pregnant women. These risks are particularly pronounced in women with medium-level education and those experiencing higher levels of stress.

From the research of the authors occurred that the pandemic exacerbated stress and anxiety among pregnant women, leading to increased substance use and subsequently higher anesthetic risks. The study underscores the importance of understanding regional risk profiles to improve anesthetic safety protocols.

Also, among the other key findings mentioned above, the authors underlined that limited access to healthcare services during the pandemic, coupled with inadequate prenatal care, contributed to increased premature births.

The study's objectives were clearly defined, focusing on identifying risk profiles, consolidating data, developing anesthetic risk models and informing public health policies.

Strengths

- Comprehensive Data Analysis - the inclusion of a large sample size strengthens the validity of the findings.

- Timely Research - the focus on the pandemic period adds relevance to the current healthcare context.

- Clear Objectives and Methodology - the structured approach to data collection and analysis provides a robust framework for the study.

Weaknesses

- Geographical Limitation - the study is region-specific, limiting the generalizability of the findings to other populations.

- Potential Bias in Self-Reporting - data on substance use and stress levels may be subject to bias due to self-reporting by participants.

In conclusion, the article provides valuable insights into the complex interplay between exogenous factors and anesthetic risk in premature births during the pandemic. It underscores the need for enhanced prenatal care, targeted public health interventions, and careful anesthetic management to improve maternal and neonatal outcomes. Future research should consider broader geographical areas and diverse populations to validate these findings further.

Overall, this study contributes significantly to the understanding of anesthetic risks in the context of premature births and highlights critical areas for healthcare improvement.

Author Response

The article investigates the influence of exogenous factors and anesthetic risks associated with premature births during the COVID-19 pandemic. The study focuses on a significant sample of 3,588 non-COVID pregnant women in Southeast Romania, among whom 297 experienced premature births. The authors analyze the impact of factors such as substance use, stress, inadequate nutrition, and socio-economic conditions on anesthetic risk and the incidence of premature birth.
The study highlights that substance use, including alcohol, tobacco, and other drugs, increases anesthetic risks in pregnant women. These risks are particularly pronounced in women with medium-level education and those experiencing higher levels of stress.
From the research of the authors occurred that the pandemic exacerbated stress and anxiety among pregnant women, leading to increased substance use and subsequently higher anesthetic risks. The study underscores the importance of understanding regional risk profiles to improve anesthetic safety protocols.
Also, among the other key findings mentioned above, the authors underlined that limited access to healthcare services during the pandemic, coupled with inadequate prenatal care, contributed to increased premature births.
The study's objectives were clearly defined, focusing on identifying risk profiles, consolidating data, developing anesthetic risk models and informing public health policies.

Authors: Thank you.

Strengths

- Comprehensive Data Analysis - the inclusion of a large sample size strengthens the validity of the findings.

- Timely Research - the focus on the pandemic period adds relevance to the current healthcare context.

- Clear Objectives and Methodology - the structured approach to data collection and analysis provides a robust framework for the study.

Authors: Thank you.

Weaknesses

- Geographical Limitation - the study is region-specific, limiting the generalizability of the findings to other populations.

- Potential Bias in Self-Reporting - data on substance use and stress levels may be subject to bias due to self-reporting by participants.

Authors: Thank you for your suggestion. We have incorporated these vulnerabilities in the conclusions section under the study's limitations and outlined future research directions accordingly.

In conclusion, the article provides valuable insights into the complex interplay between exogenous factors and anesthetic risk in premature births during the pandemic. It underscores the need for enhanced prenatal care, targeted public health interventions, and careful anesthetic management to improve maternal and neonatal outcomes. Future research should consider broader geographical areas and diverse populations to validate these findings further.

Overall, this study contributes significantly to the understanding of anesthetic risks in the context of premature births and highlights critical areas for healthcare improvement.

Authors: Thank you for your suggestions, which have helped us improve the article. We are honored by your positive feedback on the manuscript.

Reviewer 2 Report

Comments and Suggestions for Authors

1. Interesting overall 

2. Introduction is too long and the in the conclusion there is an explanation re the extensive literature review , this needs to be highlighted in the introduction.

3. Methodology has all the results so this needs to be corrected with methodology only and results in the results section.

4. One thing that needs to be clarified when you talk about anesthetic risk you need to explain more clearly ...are you meaning there is a risk from the anesthesia or do you mean the patient has a risk of needing anesthesia for childbirth ...the term anesthesia risk was not clear to after reading multiple times.

5. The correlations seem to be well supported but there are then statements with no data to support in the study such as Table 5 factors urban living and a direct influence due to pollution and toxins ; this seems to be unsupported with data , possible but a strong statement with opinion not data. The other factors also seem to have a stronger statement than the data can support as you residence , education etc have disparities but Table 5 factors need to be reconsidered.

6. Overall there is data to support your correlations but keep to the point. 

Comments on the Quality of English Language

Overal , no big issues but too long and much repetition throughout.

Author Response

  1. Interesting overall 

Authors: Thank you.

  1. Introduction is too long and the in the conclusion there is an explanation re the extensive literature review, this needs to be highlighted in the introduction.

Authors: Thank you for your suggestions. We have shortened the introduction and removed the comments on the literature review from the conclusions.

  1. Methodology has all the results so this needs to be corrected with methodology only and results in the results section.

Authors: Thank you for your suggestions. We have placed Chapter 4 of the results immediately after the presentation and analysis of the model equation.

  1. One thing that needs to be clarified when you talk about anesthetic risk you need to explain more clearly ...are you meaning there is a risk from the anesthesia or do you mean the patient has a risk of needing anesthesia for childbirth ...the term anesthesia risk was not clear to after reading multiple times.

Authors: Thank you for your suggestion. We have thoroughly reviewed all instances where anesthetic risk is analyzed and provided clarifications and observations to eliminate any confusion.

  1. The correlations seem to be well supported but there are then statements with no data to support in the study such as Table 5 factors urban living and a direct influence due to pollution and toxins; this seems to be unsupported with data, possible but a strong statement with opinion not data. The other factors also seem to have a stronger statement than the data can support as you residence, education etc have disparities but Table 5 factors need to be reconsidered.

Authors: Thank you for your suggestion. In the first column of Table 5, we included the links to the specialized literature and the data presented in Table 4 (Pearson correlations).

  1. Overall there is data to support your correlations but keep to the point. 

Authors: Thank you for your suggestion.

Comments on the Quality of English Language

Overal, no big issues but too long and much repetition throughout.

Authors: Thank you for your suggestion. We have carefully reviewed the text and removed any repetitions.

Authors: Thank you for your suggestions, which have helped us improve the article. We are honored by your positive feedback on the manuscript.